# Comparative Evaluation of Road Pricing Schemes: A Simulation Approach (Australian Perspective)

**Tariq Munir** [1,*] **, Hussein Dia** [1] **, Sajjad Shafiei** [2] **and Hadi Ghaderi** [3]

1   Department of Civil and Construction Engineering, Swinburne University of Technology, Melbourne, VIC 3122, Australia; hdia@swin.edu.au
2   Department of Computer Science and Software Engineering, Swinburne University of Technology, Melbourne, VIC 3122, Australia; sshafiei@swin.edu.au
3   Department of Management and Marketing, Swinburne University of Technology, Melbourne, VIC 3122, Australia; hghaderi@swin.edu.au
*   Correspondence: tmunir@swin.edu.au

**Abstract:** Road network pricing and congestion charging continue to be debated as efficient instruments to address traffic congestion and emissions. For cities where the schemes have not been implemented yet, the impacts of these schemes are typically evaluated using transport simulation models to understand the impacts and design effective solutions before the schemes are deployed. This paper considers a simulation approach for the city of Melbourne in Australia to investigate the potential impacts of road network pricing on reducing private vehicle travel, road congestion, and vehicle emissions. The study uses a dynamic traffic simulation model developed for Melbourne using the AIMSUN modeling tool, which was extended for modeling road user pricing and congestion charging, including considerations and formulations of distance-based, delay-based, joint-distance-and-delay-based, and cordon-based schemes under low-cost, medium-cost, and high-cost regimes. The study's contributions also include an extension of the modeling framework to include public transport options to allow for providing travelers with the option of choosing an alternative mode of transport if they do not wish to pay. A mesoscopic stochastic route choice modeling approach was adopted to examine the impact of road pricing inside a nominated charging zone within the network. The results showed it would be possible to achieve a reduction of 11% in vehicle count, a 20% reduction in travel time, a 13% reduction in emissions, and a 3% increase in travel speed within the proposed pricing zone under a high-cost pricing scenario. The results also showed a significant reduction in emissions resulting from shifting drivers who are not willing to pay the congestion charge to public transport. When 20% of car drivers shifted to public transport, carbon emissions were reduced by up to 30% and network performance improved by 45%, compared to the baseline scenario without pricing. The findings of this research provide important directions for policymakers in deciding on the type and scope of charging schemes to use and how these could reshape transportation taxation systems by moving away from taxes on vehicles through registration fees and towards user-pay taxations where travelers pay for the amount of travel they do or the pollution and emissions they are responsible for.

**Keywords:** road network pricing; congestion charging; dynamic simulation modeling; low carbon mobility; stochastic route choice experiment; dynamic transport taxation





## 1. Introduction

Road network pricing and congestion charging are two prominent traffic demand management strategies for mitigating traffic congestion and emissions, effectively promoting the efficient utilization of transportation infrastructure while generating revenue to support overall infrastructure costs. These innovative approaches shift the paradigm away from conventional revenue-raising methods like fuel taxes and vehicle registration fees

towards a more efficient, economically viable, and socially acceptable user-pay system. Numerous studies in the literature have demonstrated the substantial benefits of road pricing schemes in cities where they have been implemented or trialed [1].

For cities yet to adopt these schemes, the evaluation of the potential impacts of these schemes is typically conducted using transport simulation models [2]. These models allow for evaluating the impacts of these interventions under different traffic conditions, thus offering valuable insights for decision-makers and the public to gain a comprehensive understanding of the effects of these schemes within their specific urban contexts. While advanced congestion pricing has been successfully implemented in several global cities like Singapore, London, Stockholm, Milan, and Gothenburg, it has not been trialed or implemented in Australia yet [3]. Melbourne, in particular, presents a compelling case for testing advanced road pricing and congestion charging solutions. Infrastructure Victoria's recent report, "Victoria's Infrastructure Strategy 2021–2051", suggested that a full-scale trial of congestion pricing be undertaken in inner Melbourne [4]. This paper therefore focuses on Melbourne as a case study within the Australian context, with a specific emphasis on the Central Business District (CBD) and its periphery, as suggested in the Infrastructure Victoria report.

A number of studies in the literature have previously applied transport modeling methodologies for evaluating the impacts of proposed road pricing or congestion charging schemes [4–7]. One of the most common pricing strategies is known as cordon pricing. This approach is a key element within road network pricing strategies and is specifically designed to address traffic congestion and environmental challenges, with a primary focus on urban environments. Cordon pricing involves the imposition of tolls or charges on vehicles as they enter designated zones within a city, typically encompassing areas like the CBD or other highly congested areas. The effectiveness of cordon pricing schemes depends on various factors, including pricing thresholds, exemptions for specific vehicle categories, and the availability of viable alternative transportation options [8,9], which can be replicated in simulation environments. Notably, these schemes have been implemented worldwide and have made significant contributions to alleviating congestion, improving air quality, and enhancing urban mobility [10].

With the advancement of cutting-edge pricing surveillance technologies, the landscape of congestion pricing has evolved significantly, gaining popularity as a means to achieve greater equity and efficiency in road pricing strategies. Recent research has shifted its focus towards the development of more sophisticated road pricing methods, with particular emphasis on distance-based and time-based approaches [11]. In these innovative frameworks, vehicles are subjected to charges that are intricately calculated based on both the distance they travel and the duration of their journeys through congested regions. This approach represents a notable departure from traditional flat-rate fees, as it seeks to create a more nuanced and fair pricing structure that reflects the actual usage of road infrastructure. By considering distance, time, or their combination, this approach aims to incentivize more efficient travel behaviors and reduce congestion in urban areas, ultimately contributing to improved traffic management and environmental outcomes [12].

This study utilizes a dynamic traffic simulation model for Melbourne that adopts a mesoscopic stochastic route choice modeling methodology. The study examines the influence of road pricing within a designated charging zone within the network, testing various state-of-the-art and state-of-the-practice road pricing schemes, including distance-based, time-based, and combinations thereof, each with different pricing structures ranging from low-cost to medium-cost and high-cost options.

This paper is organized as follows: Section 2 provides an overview of simulation studies that have evaluated road pricing schemes. Section 3 presents the methodology employed in this study, including the sub-network in Melbourne, offering insights into the setup and implementation details. Section 4 presents the numerical results, accompanied by a detailed analysis of the outcomes and performance metrics. Lastly, Section 5 encapsulates

the research findings and offers concluding remarks that succinctly summarize the key contributions of the work and potential directions for future research extensions.

## 2. Background Literature on Road Pricing Simulation Studies

A number of studies in the literature have previously applied transport modeling methodologies for evaluating a range of pricing solutions, including cordon pricing [13], distance-based pricing [14], parking pricing [6], road pricing in relation to parking pricing [15], freight charging [16], charging based on emissions [17], and combinations of these approaches [2]. A summary of these studies, which are relevant to the scope of work described in this paper, is provided next and summarized in Table 1.

### 2.1. Cordon Pricing Simulation Studies

In a cordon-based pricing study in Isfahan city, Iran, the authors used a simulation model developed in TransCAD supplemented with a stated preference and revealed preference survey data collected from 480 car users in the city center. The modeling results revealed that 74% of users stopped using their private cars to enter the city under a USD 1.43 cordon-based charge, and 32% were not willing to pay the charge. Car users shifted their mode of travel, and 30% started using taxis, 23% buses, and 20% walking, resulting in reduced air pollution and a 2–3% increase in travel speeds [10]. Gu and Waller [18] presented a solution to two-toll-level problems in their study for Melbourne, Australia, which included travel time optimization and the uneven distribution of congestion in a pricing cordon using a surrogate-based optimization method (response surface method or meta modeling), which was developed in AIMSUN. The results showed that the two optimal toll level problem solutions reduced average travel time by 29.5% and 21.6%, respectively, in the pricing cordon. When the drivers' heterogeneous behavior was considered, the congestion was further reduced, and the pricing resulted in smoothing traffic flow, reducing travel time and the total time saved in traveling throughout the cordon network [18].

In another study for New York City [19], a traffic simulation approach based on the TransCAD simulator was undertaken to investigate driver behavior and vehicle emissions impacts resulting from a proposed cordon-based pricing scheme. The model simulated a number of congestion pricing scenarios that included charges of USD 5, 10, 15, and 20 for entry into the city. The results revealed a considerable decrease in trips to and from the CBD, a 6% increase in public transport usage, a reduction in single-occupancy vehicles and taxis by 30% and 40%, respectively, and a reduction in emissions and air pollutants, especially particulate matter, by 17.5% in the USD 20 pricing scenario. Travel delays for all modes of transport were reduced by 32%, and vehicle kilometers traveled (VKT) were also reduced by around 14%.

In Zurich, the traffic simulator MATSim was used to model a dynamic cordon pricing scheme to determine an optimal charge for an urban congested network. A cordon-area of 1 km radius was proposed to be in operation from 7:30 a.m. to 8:00 p.m., with dynamic charging ranging from EUR 1 to EUR 4. The results showed that traffic congestion reduced in high charging scenarios, while the traffic density in neighboring areas increased slightly during these periods. The study also found that drivers using their vehicles for work-related activities were less likely to shift their departure times even in higher charging scenarios, as compared to those going for non-work or leisure activities [20].

Studies in Singapore and Boston used a traffic simulation approach to explore the impacts of dynamic distance-based congestion charging schemes using the simulation tool DynaMIT2.0. The results revealed that social welfare would be best achieved in terms of the pricing scheme, in which charging zones were defined based on the marginal cost. The results showed that network dynamics were significantly improved for time-based and cordon-based schemes compared to existing non-pricing scenarios [11].

## 2.2. Pricing and Reward Strategies

A preference survey study conducted in Beijing, China, explored the efficiency of congestion pricing and reward strategies during morning peak hours. The study found that habitual travelers were more likely to shift their modes to other sustainable modes, e.g., walking, e-bikes, and public transport, under a pricing scheme than less habitual commuters. However, less habitual travelers preferred to shift their modes under a reward scheme. The amount of this monetary award was designed to cover the full or partial travel costs of the participants if they wished to use public transport. For a train ticket starting at 3 yuan, travelers would receive a 50% discount if their monthly travel cost exceeded 150 yuan [21].

A study was carried out using a mathematical modeling approach in Winnipeg, Canada, to determine the usefulness of a subsidy scheme in which drivers on some (congested) roads will be charged but also rewarded with a bonus amount (subsidy) for using alternate (less congested) roads at the same time. It was concluded that the subsidy attracts travelers to divert to alternate routes. The analysis revealed that the traffic volume decreased notably in the tolled scenario [22].

In another mathematical modeling study that examined the impacts of congestion pricing combined with the presence of dedicated bus lanes and public transport subsidies, users were provided with the option to choose cars, buses, and bikes. The results showed that the public transport subsidy scenario had notably increased public transport ridership by attracting car users. Congestion pricing in the presence of public transport subsidies performed even better and attracted more people from cars to buses as compared to the public transport subsidy alone. The reduction in traffic congestion improved the overall network performance, leading to an increase in average travel speeds. The provision of a dedicated bus lane scenario achieved a threefold increase in bus speeds. The authors argued that among all these scenarios, the bus lane introduction was the most effective policy and was able to achieve maximum social and consumer surplus. The study also calculated an optimal value of congestion charge, which was around USD 0.18 [23]. In a similar study in the context of passenger and freight vehicles, it was concluded that the revenue generated from congestion pricing can be redistributed among the users of this policy as an incentive, which can lead to enhancing social welfare [24].

## 2.3. Combination of Multiple Strategies

In a simulation study that explored the impact on mode choice and parking location choice in the presence of cordon pricing and a parking pricing policy for Mashhad CBD, Iran also considered cordon pricing. A stated preference (SP) and revealed preference (RP) survey was conducted at parking locations to get public opinion about the pricing policies. It was concluded that it is more likely for travelers going towards the CBD to shift to modes other than cars under higher parking and cordon tolls. It was found that if cruise time for parking and travel delays inside the cordon area caused delays, then travelers would avoid car modes and would prefer to take a taxi. Respondents were likely to opt for public transport for travel to CBD if the services improved [25].

Another modeling study compared the impacts of different congestion pricing strategies on urban land use in monocentric and polycentric city scenarios. Three tolling scenarios that included Marginal Cost Pricing (MCP), the Vehicle Mile Traveled (VMT) toll, and the cordon toll were considered. It was concluded that the congestion pricing policy contributed to creating a more compact city. VMT and cordon toll created higher social welfare for households and a more compact urban form, while MCP allowed firms and households to choose a longer distance to avoid traffic congestion. All three CP strategies provided a reduction of 10% in the daily commute shift of 16% jobs from CBD to other suburban areas [26].

Finally, a study conducted in Zurich, Switzerland, analyzed the impacts of congestion pricing and parking charges when provided in combination with a park-and-ride facility.

The cruising time for parking was reduced by 30%, and the average delay was reduced to 50% due to drivers' decision to use the park-and-ride facility for entering the CBD [27].

**Table 1.** Background literature on road pricing modeling studies.

| Reference | City and Country | Methodology | Pricing Strategy | Benefits/Results |
|---|---|---|---|---|
| **Cordon Pricing Simulation Studies** | | | | |
| [10] | Isfahan, Iran | TransCAD simulation model | USD 1.43 cordon-based charge | 74% of users stopped using their private cars |
| [19] | New York, USA | TransCAD simulation model | Cordon-based pricing in USD 5, 10, 15, and 20 scenarios | 6% increase in public transport usage, reduction in car and taxi flow by 30% and 40%, reduction in emissions by 17.5% |
| [20] | Zurich, Switzerland | MATSim traffic simulator | Dynamic cordon pricing: EUR 1 to EUR 4 | Traffic congestion is reduced in high-cost charging scenarios |
| [11] | Singapore and Boston | DynaMIT2.0 | Dynamic, distance-based pricing | Network dynamics were significantly improved |
| [18] | Melbourne, Australia | Surrogate-based optimization using AIMSUN | Two optimal toll-level problem solutions | Average travel time was reduced by 29.5% and 21.6%, respectively |
| **Pricing and Reward Strategies** | | | | |
| [22] | Winnipeg, Canada | Mathematical modelling | Congestion pricing and subsidy scheme | The traffic volume decreased notably in the pricing scenario as the subsidy diverted travelers to alternate, non-priced routes |
| [23] | Hypothetical network | Mathematical modelling | Congestion pricing combined with dedicated bus lanes | Bus speeds improved three times, resulting in more people shifting from cars to buses |
| [24] | - | Mathematical modelling | Congestion pricing with revenue redistribution | Enhanced social welfare |
| **Combination of Multiple Strategies** | | | | |
| [26] | Hypothetical network | Simulation modelling | Marginal Cost Pricing, Distance, and cordon pricing | All strategies provided a reduction of 10% in daily commutes and a shift of 16% of jobs from the CBD to suburban areas |
| [27] | Zurich, Switzerland | Simulation modelling | Congestion pricing and parking charges with a park-and-ride facility | The cruising time for parking was reduced by 30%, with the average delay reduced by 50% |

## 3. Methodology and Modeling Framework

This paper includes findings from research that aims to demonstrate the effectiveness of road network pricing as a travel demand management strategy. The research focused specifically on the City of Melbourne, Australia, and considered Melbourne CBD and nearby areas surrounded by CityLink, the West Gate Freeway, Punt Road, and Alexandra Parade. This area was selected for the simulation study because it had been proposed as a target congestion charging area in the Infrastructure Victoria Strategy 2021–2051. The research also investigates the role of different types of road network pricing in reducing private vehicle travel, road congestion, and environmental emissions from passenger vehicles in urban and high-traffic areas and provides a comparative evaluation of their impacts.

A simulation approach was selected because it provides a virtual testbed where different scenarios can be tested and would serve as an evidence base for any future work

that looks into the deployment of road network pricing in Melbourne. Traffic simulation tools [28] are capable of modeling both vehicles and passengers from their origins to specific destinations. They allow modeling to test different travel management policies and scenarios to evaluate their impacts before their implementation in the real world. In addition to being economical and socially safe, simulation tools are also effective for model development, analysis, and evaluation. Researchers and policymakers have greatly benefited from these tools to carry out studies on large-scale transportation networks to assess their performance. There are many traffic simulation software packages available, including VISUM, SATURN, TRACKS, DYNAMIT, DYNAMEQ, AIMSUN, etc., providing opportunities to evaluate and compare the traffic network dynamics [29].

### 3.1. AIMSUN Traffic Simulator

In this study, the well-known traffic simulation tool AIMSUN Next 20 is used. AIMSUN has been extensively studied and used successfully [30] and was selected for this study because the underlying large-scale model to be used for modeling road pricing scenarios for Melbourne was also developed in AIMSUN. In general, traffic simulation tools are capable of modeling both vehicles and passengers from their origins to their destinations [28]. The AIMSUN model works by tracking each agent (vehicle-driver combination) and recording information related to their performance throughout the simulation period. The model applies a variety of mathematical models and traffic flow theories to simulate passenger and driver behaviors and also accounts for the evolution and dynamics of traffic in the network. Model development relies on establishing a set of calibrated and validated parameters that result in outputs that replicate reality within a pre-specified degree of accuracy.

At a broad level, AIMSUN includes a hierarchy of levels (macroscopic, mesoscopic, and microscopic) that can be used to test the pricing schemes. The multi-layer modeling approach provides options for testing innovative pricing solutions at a number of levels, depending on the desired objectives. For example, macro modeling is a strategic tool used for demand forecasting and for exploring the impacts of pricing at a regional or metropolitan level, such as pricing solutions that focus on demand management as an alternative to building additional infrastructure.

Meso-modeling, which was used in this research, provides a dynamic and operational approach and is mainly used for estimating time-dependent itinerary choices in a congested network based on user perceptions of generalized costs. The last and most detailed layer, micro-modeling, provides tools to evaluate in greater detail the impact on travel time and operational efficiency of network areas where a high frequency of interactions between vehicles, public transport, cyclists, and pedestrians occurs. Together with the provision of data feeds, this tool allows for a holistic evaluation of mobility solutions at different levels of complexity and geographical coverage.

### 3.2. Dynamel–Dynamic Model for Melbourne

This research is based on an agent-based modeling framework that was developed using the traffic simulation tool AIMSUN Next. In this study, the simulation tool AIMSUN was used [30–32]. The simulation model tracks each agent (car-driver combination) and records information related to them throughout the simulation period. The model applies a variety of mathematical methods and traffic flow theory to simulate passenger and driver behavior and traffic phenomena in the network. This relies on calibrated and validated parameters and detailed and varied input data about the transportation system capable of generating an extensive array of output statistics. In the model, traffic flow is represented at the microscopic level. Simulation of vehicles on the road is based on car-following, lane-changing, and gap-acceptance algorithms. Featured models include Gibbs, Weidmann, and Fritzsche. The stochastic nature of the travelers in a traffic simulation model requires that the model iterate the simulation to produce realistic statistical outputs. In addition, vehicle demand is characterized by an origin-destination matrix of vehicle trips comprising origin, destination, and departure time. Agents or drivers can dynamically

choose the best route and mode in real-time. The simulation results provide a set of statistics, including information on travel speed, delay, flow, amount of fuel consumed, and production of emissions [33,34]. A more detailed coverage of the AIMSUN simulation tool can be found in [28].

The "Dynamel" model for Melbourne, which is a dynamic traffic assignment model, is used in this study. This is a mesoscopic model that has been extensively used to study the evolution and propagation of the dynamics of traffic congestion. It describes the dynamics of traffic congestion with time-based network conditions and allows users to develop network planning and transportation decision support systems. It is an open-access model available for public use that was adapted to undertake this research. The Melbourne model schematic is shown in Figure 1.

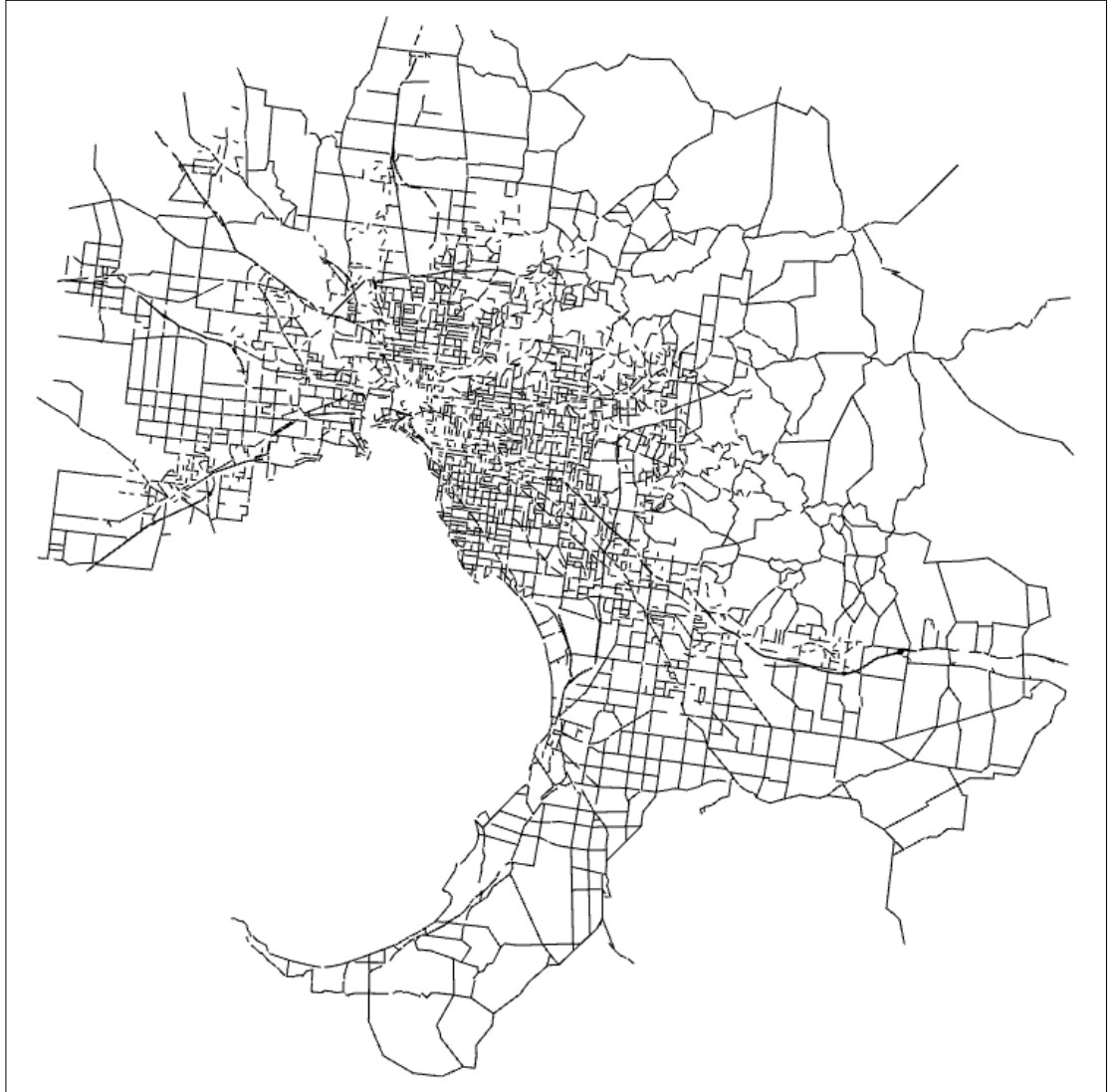

**Figure 1.** Schematic of the Melbourne Dynamel Network.

The model uses a comprehensive travel demand for almost 2.1 million commuters in a 4-h morning peak. This demand corresponds to travel information for 120,000 daily trips in Melbourne, including 29,576 trips that are made using private vehicles during the morning peak [35]. The readers can refer to [5] for a more detailed coverage of the Dynamel model. The model has been extensively calibrated based on real-world data collected from the Victorian Integrated Survey of Travel Activity (VISTA). The model was previously used in a number of studies, and its performance and predictions have been shown to provide a valid reflection of ground reality [36].

*3.3. Road Pricing and Route Choice*

A key consideration in the simulation of the impacts of road network pricing schemes is the formulation of route choice models under road pricing [5,15].

The network, denoted as $G = (N, A)$, undergoes an initial partitioning into pricing and non-pricing regions, each denoted as $G_1 = (N_1, A_1)$ and $G_2 = (N_2, A_2)$, with the condition that $G_1$ and $G_2$ equals the original network G.

This study posits that the toll charges, contingent on distance traveled and travel time, exhibit a linear relationship within each region, as expressed by the equations:

$$\varnothing(d) = \alpha_1 \cdot d_1 + \alpha_2 \cdot d_2, \quad d = d_1 + d_2 \tag{1}$$

$$Y(t) = \beta_1 \cdot t_1 + \beta_2 \cdot t_2, \quad t = t_1 + t_2 \tag{2}$$

where $\alpha_1$ and $\beta_1$ are positive distance and delay toll rates in $G_1$, respectively ($\alpha_2, \beta_2 = 0$). The notation "*rs*" represents the pair of origin and destination ("*rs*"), while "$K^{rs}$" denotes the set of paths connecting origin-destination pairs "*rs*". Each path "*k*", where $k \in K^{rs}$, is one of the possible paths connecting origin "*r*" to destination "*s*". The notation "*a*" is used to denote a link within path "k". Furthermore, "$\xi$" is employed to represent the proportion of the commonality factor. Let $l_k^{rs}$ and $\tau_k^{rs}$ denote the total distance and time traveled for route *k* between the origin–destination *r-s* pairs for the path $k \in K^{rs}$ :

$$l_k^{rs} = \sum_{a \in A_1} l_a \cdot \delta_{a,k}^{rs} + \sum_{a \in A_2} l_a \cdot \delta_{a,k}^{rs}, \quad k \in K^{rs}, \quad rs \in RS \tag{3}$$

$$\tau_k^{rs} = \sum_{a \in A_1} \tau_a \cdot \delta_{a,k}^{rs} + \sum_{a \in A_2} \tau_a \cdot \delta_{a,k}^{rs}, \quad k \in K^{rs}, \quad rs \in RS \tag{4}$$

where $\delta_{a,k}^{rs} = 1$ if path $k \in K^{rs}$ contains link *a*, and $\delta_{a,k}^{rs} = 0$ otherwise. By merging Equations (1)–(4), we derive the distance- and time-based toll for the path denoted as $k \in K^{rs}$ as follows:

$$\varnothing(l_k^{rs}) = \alpha_1 \cdot \sum_{a \in A_1} l_a \cdot \delta_{a,k}^{rs} \tag{5}$$

$$Y(\tau_k^{rs}) = \beta_1 \cdot \sum_{a \in A_1} \tau_a \cdot \delta_{a,k}^{rs} \tag{6}$$

An investigation was conducted on this network to assess the ramifications of various pricing strategies on traffic dynamics. The study explored the following scenarios based on a previous study [5]:

1. Distance-Based Pricing: In this approach, vehicles incur charges based on the distance they travel within a designated area, computed on a per-kilometer basis.
2. Delay-Based Pricing: Under this strategy, fees are imposed on vehicles depending on the delays experienced during their journeys.
3. Joint Distance and Delay-Based Pricing: In this particular scenario, both distance-based and delay-based pricing mechanisms are simultaneously applied.
4. The route cost function for route *k* between origin and destination *rs* is expressed as:

$$G(g_k^{rs}) = T_k^{rs} + \omega_1 . \varnothing(l_k^{rs}) + \omega_2 . Y(\tau_k^{rs}) \tag{7}$$

For the distance-based scheme, $\omega_1 = 1$, $\omega_2 = 0$, delay-based $\omega_1 = 1$, $\omega_2 = 0$ and for the joint pricing $\omega_1 = 1$, $\omega_2 = 1$. The general cost function allows for the computation of the probability associated with selecting a specific route. In this investigation, the C-logit model [37] is employed, which enhances the conventional logit model by incorporating a "commonality factor".

The C-logit model aims to overcome the assumption of identical and independent distribution (IID), which is often invalid for route choice problems within extensive road networks. This model estimates the probabilities of paths that share common links by introducing the "commonality factor". This factor exhibits an inverse relationship with path k's degree of independence from other paths, assuming a value of zero when no other path shared links with path k [37]. The commonality factor is mathematically defined as:

$$CF_k = \ln\left(1 + \sum_{h \neq k} \frac{z_{hk}}{\sqrt{z_h z_k}}\right) \tag{8}$$

In this equation, $z_h$ and $z_k$ represent cumulative values of the cost attribute over the links in paths h and k, respectively, while $z_{hk}$ signifies cumulative cost-attribute values over the links shared by both paths. The attribute CF exhibits greater value for paths where shared links contribute significantly to their overall cost or length. Consequently, the C-logit model effectively reduces the probability of selecting heavily overlapped paths, rendering it a more realistic route choice model.

With $CF_k$ as a parameter, the C-logit model's formulation is as follows:

$$p(k) = \frac{e^{(-G(g_k^{rs}) - \xi . CF_k)}}{\sum_{h \in K^{rs}} e^{(-G(g_k^{rs}) - \xi . CF_k)}} \tag{9}$$

Here, p(k) represents the probability associated with selecting path k, $G(g_k^{rs})$ denotes the general cost attributed to path k, $K^{rs}$ denotes the set of paths connecting origin-destination pairs *rs*, and $\xi$ is used to denote the proportion of the commonality factor, which required a proper definition within the model [36].

### 3.4. Study Area

As discussed earlier, congestion pricing has been implemented in many cities around the world, e.g., Singapore, London, Stockholm, Milan, and Gothenburg, among others. It has provided promising benefits in terms of reduced congestion and improved vehicle speeds. However, it has neither been trialed nor implemented in Australia. Melbourne, being one of the world's most livable cities, serves as a good example of where road pricing and congestion charging can be tested. This study has considered a large area surrounding the Melbourne CBD, which extends to CityLink, the West Gate Freeway, Punt Road, and Alexandra Parade. The key consideration in selecting this study area is that it included multi-modal interactions (e.g., private passenger vehicles, trains, trams, buses, etc.), and it also had adequate alternative non-priced routes for travelers' who do not wish to pay the road user charge, thus providing them with access to alternative non-priced routes as well as alternative modes of transport, such as public transport. The selected study area also coincides with the cordon-priced congestion charging zone proposed by Infrastructure Victoria, the state's peak infrastructure body, which has been advocating the use of congestion charging for the city.

This paper therefore considers Melbourne as a case in an Australian context, with a focus on the Central Business District and periphery. In a recently published report (Victoria's infrastructure strategy 2021–2051), by peak body Infrastructure Victoria, a proposal for a full-scale trial of congestion pricing in inner Melbourne suggested the following areas for road pricing (Figure 2).

That same area has been selected as the study area in this research for testing the impact of road pricing in Dynamel. In Figure 3, the outer boundary in the diagram represents the simulation area, and the inner boundary is the pricing zone. The outer rectangular boundary is the subnetwork extracted from the larger network and is comprised of 6717 sections and 2969 nodes with 1604 km of total section length of roads. The inner area, which is bounded by CityLink, the West Gate Freeway, Punt Road, and Alexandra Parade, is selected as a pricing cordon to test various scenarios for road pricing. This inner area is comprised of a total of 1713 sections of road. The travel demand within the area was generated using

dynamic traversal for the subnetwork for the 4-h morning peak from 6 a.m. to 10 a.m. The simulated demand includes 558,232 vehicles during this period.

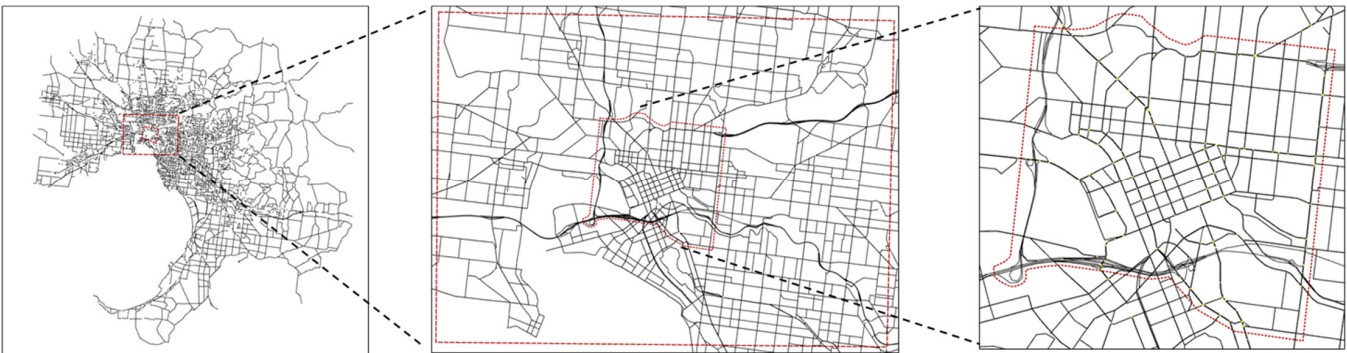

**Figure 2.** Study area—Melbourne CBD and surrounding areas.

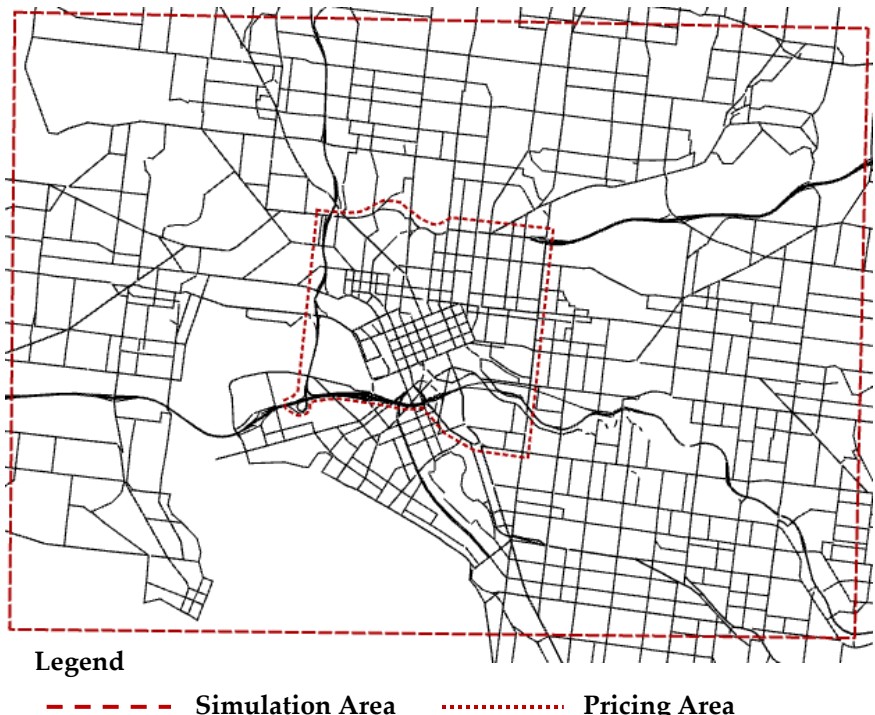

**Legend**

– – – – **Simulation Area** ·············· **Pricing Area**

**Figure 3.** Simulation study area—Melbourne.

*3.5. Simulated Pricing Schemes*

Four different pricing strategies were tested on this network to evaluate the impact of road pricing on traffic dynamics. The following scenarios were examined:

- No pricing scenario (this serves as a baseline for comparison purposes)
- Distance-based pricing
- Delay-based pricing
- Joint-distance and delay-based pricing

In distance-based pricing schemes, a charge is applied to vehicles that move within an area on a per-kilometer basis. The charge can be fixed or variable according to traffic or emissions conditions. In Germany, for example, a distance-based charge was implemented on high-emitting vehicles at EUR 0.14 per kilometer. The delay-based pricing charging scheme applies a charge to vehicles depending on the travel delay they make within the pricing zone. A joint scheme was also tested, comprising both distance- and delay-based pricing, which charges vehicles simultaneously considering both indicators, i.e., distance

and delay. The outputs were then compared with the no-pricing scenario (baseline current conditions without pricing or congestion charges).

Setting up these scenarios in AIMSUN required the creation of a subnetwork, which was extracted from the main model, and an estimate of demand for that specific part, which was achieved using a dynamic traversal that produced exactly the same 16 demand matrices as the whole network in 15-min intervals. The travel demand definition comprised Origin–Destination (OD) matrices, which specify the number of people and vehicles traveling from each origin to each destination. The Dynamel model used in this study comprised demand for the morning peak from 6–10 a.m. The travel demand data in Dynamel was obtained from the Victoria Integrated Transport Model (VITM), comprising a 2974 OD matrix. The Victorian Integrated Survey of Travel and Activity (VISTA) data was also used, incorporating information about the different departure time choice patterns between OD pairs. After setting the demand for travel, the cost functions required to implement pricing were developed. Inside the inner pricing cordon, initial (baseline) and dynamic cost functions were set up for the priced links. Similarly, initial and dynamic cost functions for non-pricing links were coded for links outside the cordon. Dynamic scenarios were then created within the subnetwork, including definitions of traffic demand and control plans. A pricing Application Programming Interface (API), developed in Python code, was used for the implementation of pricing on the specified links. This API starts the pricing at 8:00 a.m., i.e., after two hours of running the simulation. Under this scenario, mesoscopic Stochastic Route Choice (SRC) experiments are then defined. The stochastic nature of how traffic is generated and distributed across the transport network model requires that the model iterate the simulation to produce realistic statistical outputs. A dynamic simulation approach was used in this study, meaning that the route cost or travel times are dynamically updated in every simulation step to reflect the prevailing traffic conditions. The simulator therefore dynamically routes vehicles according to the shortest travel time to their destinations.

To account for the randomness of results and the stochastic nature of traffic, a number of replications are created, and the average results are considered for all replications.

The pricing scenarios were then defined based on the cost values for each pricing scheme. Three scenarios were specified, namely low-cost (AUD 0.1 per kilometer), medium-cost (AUD 0.25 per kilometer), and high-cost (AUD 0.5 per kilometer). The cost for the congestion charging scenarios was set based on studies from the literature where the value of time was reported as AUD 20 per hour [5]. Other studies reported different values; e.g., in a congestion pricing study based on user heterogeneity, six income level groups were defined based on their respective values of travel time, including 6, 12, 19, 27, 36, and 49 Swiss Francs (CHF) per hour, where CHF is equivalent to USD 1 [38]. The advantage of using a traffic simulation tool is that it allows for these values, modeling, and pricing parameters to be changed and the impacts evaluated.

### 3.5.1. Low-Cost Scenario

In this scenario, the distance-based pricing cost was set to be AUD 0.1 per kilometer. This means that travelers would be charged 10 cents for each kilometer they have traveled in the pricing cordon. The time-based pricing was defined as AUD 10 per hour in this scenario. This means that travelers would be charged 10 dollars per hour they spent in the pricing zone. For comparison, the charging reported in other scenarios is presented next. In a congestion charging study, the amount of charge was set at a USD 500 yearly fee in the restricted traffic zone, which is equivalent to USD 1.36 per day [39]. In another cordon pricing scheme study, the charge amount was defined as 3.2 yuan/trip, which is representative of USD 0.5 [40]. In a time-dependent pricing scheme, the optimal charge amount defined by the simulator was between USD 0.5 and USD 0.3 per vehicle per hour in the evening peak [14].

### 3.5.2. Medium-Cost Scenario

In this scenario, the distance-based pricing cost was set to be AUD 0.25 per kilometer. Similarly, this means travelers would be charged 25 cents for each kilometer they have traveled in the pricing cordon. The delay-based pricing was defined as AUD 20 per hour in this scenario, meaning, as described above, that travelers would be charged 20 dollars per hour they spent in the pricing zone. For comparison, a previous paper in a dynamic cordon pricing scheme considered the charge amount EUR 2–10 during different times of the day (EUR 1 is equal to USD 1 [20]). In another area-based pricing strategy, researchers chose the charge amount of 5 CHF to be implemented (where CHF is equivalent to USD 1 [38]).

### 3.5.3. High-Cost Scenario

In this scenario, the distance-based pricing cost was set to be AUD 0.5 per kilometer, and the delay-based pricing was defined as AUD 30 per hour. For comparison, a study that analyzed user-based charges and subsidy schemes applied a charge amount of USD 1.19 for the shortest one-way travel route (37 min long), while others received a subsidy of USD 0.65 and USD 1.37 for taking longer routes of 40 min and 43 min, respectively [41]. In a study of the impacts of congestion pricing and reward strategies on travelers' commuting, charge amounts were set to 5 yuan, 15 yuan, and 25 yuan in three different scenarios, where USD 1 is equal to 7 yuan [21]. A summary of the charging amounts used for the different pricing strategies implemented in this paper is provided in Table 2.

**Table 2.** Charge amounts in different pricing scenarios.

| Scenario | Scenario Name | Distance-Based Charge Amount (Per Km) | Delay-Based Charge Amount (Per Hour) |
|---|---|---|---|
| 1 | High cost | AUD 0.50 | AUD 30 |
| 2 | Medium cost | AUD 0.25 | AUD 20 |
| 3 | Low cost | AUD 0.10 | AUD 10 |

## 4. Results

All simulation results generated by AIMSUN reflected improvements in traffic performance as a result of implementing road pricing, compared to the baseline no-pricing scenario. The outputs include vehicle count, travel time, queue length, emissions levels, and travel speeds.

### 4.1. Vehicle Count

The vehicle count in a network refers to the total number of vehicles, which is a measure of the performance of each scenario. A successful congestion charging or road pricing scenario would have demand management effects and result in a lower number of vehicles in the network compared to the baseline non-pricing scenarios. In this study, the simulation starts at 6:00 a.m., but the pricing starts at 8:00 a.m., i.e., after two hours from the simulation starting period. The results are shown in Figure 4. In the low-cost scenario, the distance-based pricing scenario was found to perform most effectively, showing the minimum vehicle count inside the pricing cordon. This means that most travelers diverted to other routes that were not priced, and hence the vehicle count was reduced within the pricing cordon.

In the low-cost scenario, the vehicle count was low in all pricing schemes as compared to the non-pricing Figure 4a, resulting in improved network performance. However, after 9:30 a.m., there is a sharp decrease in vehicles in joint-distance- and delay-based schemes. In this scenario, the charge is quite high because it is based on both indicators, i.e., distance traveled and time spent in the proposed pricing zone. Hence, drivers who have flexible time and do not need to be at their destinations in strict time frames can alter their usual route, which reduces traffic congestion [21].

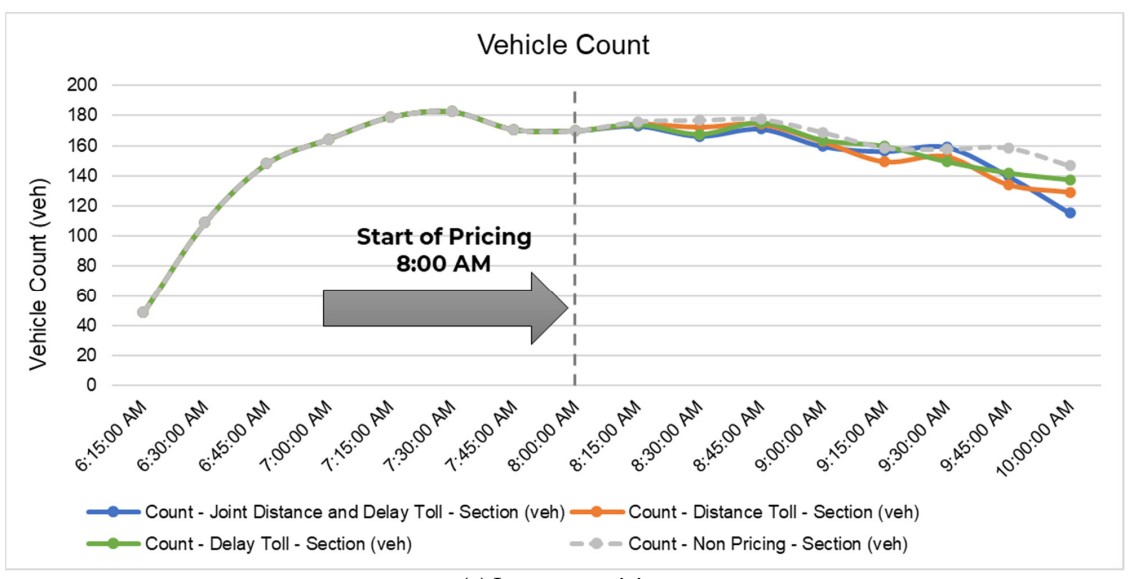

(**a**) Low-cost pricing

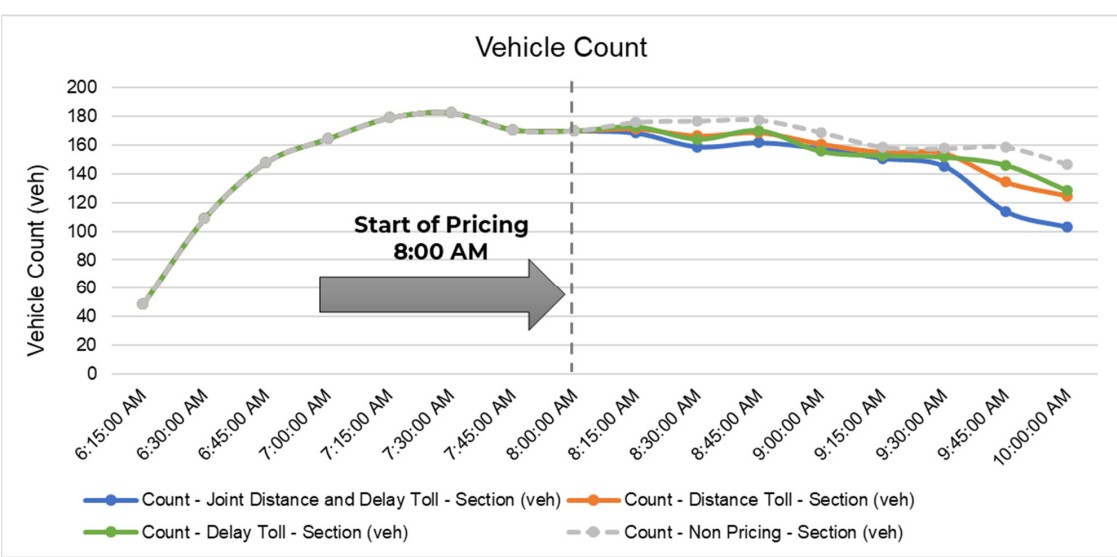

(**b**) Medium-cost pricing

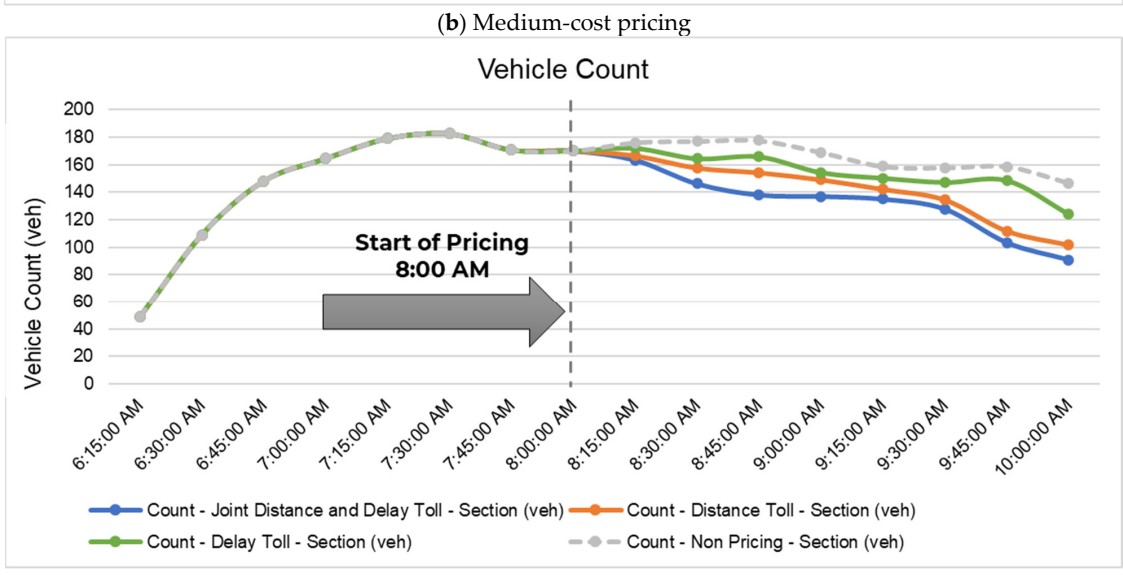

(**c**) High-cost pricing

**Figure 4.** Simulation results of the pricing zone under different pricing scenarios.

In the medium-cost scenario, vehicle counts further decreased in all pricing schemes as compared to the non-pricing scenario (Figure 4b). The network's performance and traffic flow had further improved. However, the joint-distance and delay-based schemes performed the best among all scenarios tested. In this scheme, the vehicle count is the lowest as compared to the other schemes. The simulation results for the high-cost pricing scenario, on the other hand, show a notable reduction in the vehicle count. In all the pricing scenarios, there is a clear reduction in vehicle count. The reduction can be observed in descending order, starting with delay-based pricing, followed by time-based pricing, and joint distance and delay-based pricing. Hence, the joint scheme performed best among all scenarios in Figure 4c. These results revealed that it is possible to achieve a reduction of up to 11% in vehicle count in a high-cost scenario. This is in line with the literature, e.g., a congestion pricing study achieved a 6% increase in public transport ridership, leading to a 40% decrease in single-occupancy vehicles under a USD 20 charge scenario [19]. In a cordon pricing scheme, other researchers achieved only a 35% reduction in vehicular traffic under a EUR 5 toll scenario [42].

### 4.2. Vehicle Count vs. Maximum Queue Length

In transport simulation studies, another important network performance parameter is the maximum queue length formed by vehicles, which is indicative of network congestion [43]. In this analysis, the authors calculated the sum of the maximum queue length and compared the simulation results with the vehicle count (Figure 5) for the high-cost pricing scenario.

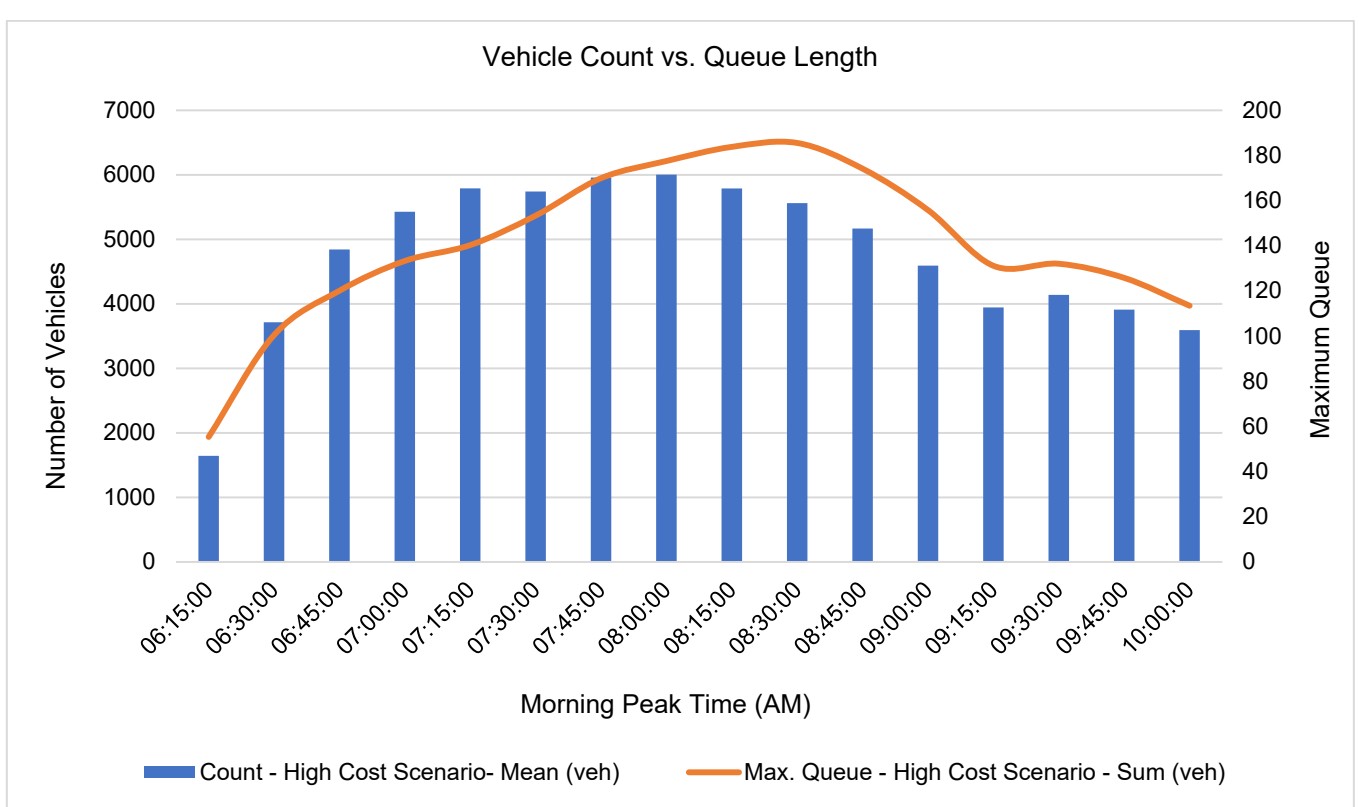

**Figure 5.** Vehicle count vs. max. queue length.

The findings show that queue length is associated with vehicle count at all times during the simulation period. Before the start of pricing at 8:00 a.m., the queue length is lower than the vehicle counts, which means that the network is still not congested. After the start of pricing, the queue length can be seen to be somewhat higher than the count, specifically at times between 8:30 and 8:45 a.m., which is also in line with the findings in other studies. This means that the travelers choose to take the pricing routes to avoid the

delay, which leads to an increase in queue length. However, at the very next moment, i.e., 9:00 a.m., it again comes down and corresponds more closely with the vehicle count.

A link-by-link traffic analysis of the network was also performed, revealing a clear difference in vehicle counts in both pricing and non-pricing scenarios. In Figure 6, the red-colored links represent a high vehicle count, i.e., traffic congestion; the yellow and orange colors represent moderate congestion; and the green color represents no traffic congestion and a smooth flow of traffic. The results showed that traffic conditions improved inside the network but worsened outside the pricing zone. This travel behavior can be interpreted in terms of travelers' route-choice decisions. Travelers who were not willing to pay the congestion charge preferred to take other non-priced and longer routes to avoid the fee, which resulted in increased congestion outside the pricing area. This expected behavior is in line with the findings of many recent studies [22,43–45] and also points to the importance of the need to ensure adequate levels of public transport services before considering the implementation of road network pricing to ensure that travelers who do not want to pay the congestion charge have alternative travel options to choose from.

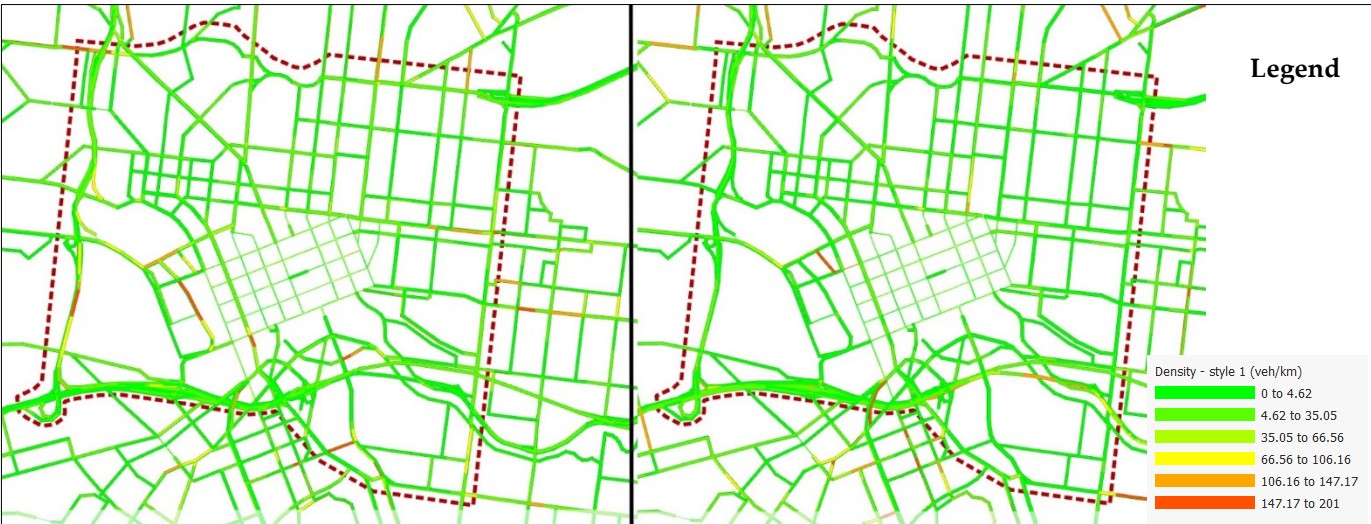

**Figure 6.** Simulated vehicle counts within the pricing zone for (**right**) pricing and (**left**) non-pricing scenarios.

*4.3. Total Travel Time*

Total travel time is the total time taken by vehicles to cover the distance from their origin to their desired destination. This is an important measure of network performance, where reduced travel time reflects improved network conditions. In this experiment, the impacts of road pricing on total travel time within the pricing zone were evaluated first. When the pricing starts at 8 a.m., there is a notable difference in vehicle travel time in all the pricing scenarios. In the low-cost scenario, the total travel time goes up and down for the whole pricing duration (8–10 a.m. for all pricing scenarios), but the overall vehicle count is low in the distance-based and joint distance- and time-based pricing scenarios. The results in Figure 7 show that the distance charge performs better consistently throughout the pricing duration. However, the joint distance–delay-based charge has the highest benefits at 9:15 a.m., which may be due to the fact that in this scenario, most travelers had decided to take the priced route because of the low-priced road user charge; hence, the network got congested and travel times increased. In the medium-cost scenario, it can be clearly seen that there are reductions in travel time in all pricing scenarios. As there is an increase in the amount of road user charge, drivers start to avoid the charging zones and take alternative free routes, as found by other similar studies [26,44]. Furthermore, as the charge amount got higher in the high-cost scenario, the total travel time was further reduced in all pricing scenarios. The difference in reductions is much higher for the high-price scenario compared to the medium-cost scenario. Therefore, the higher charge can further lower the

amount of travel within the pricing zone where travelers choose to drive on alternate free routes. Further examination of the results showed that the joint-distance and delay-based schemes performed better compared to other schemes. The simulation results achieved a reduction of up to 20% in total travel time in the high-cost scenario, which is much better than findings in studies conducted in Portugal, which achieved a reduction of 6% [46]. In other scenarios, there was a 4% reduction in the low-cost scenario and a 15% reduction in the medium-cost scenario.

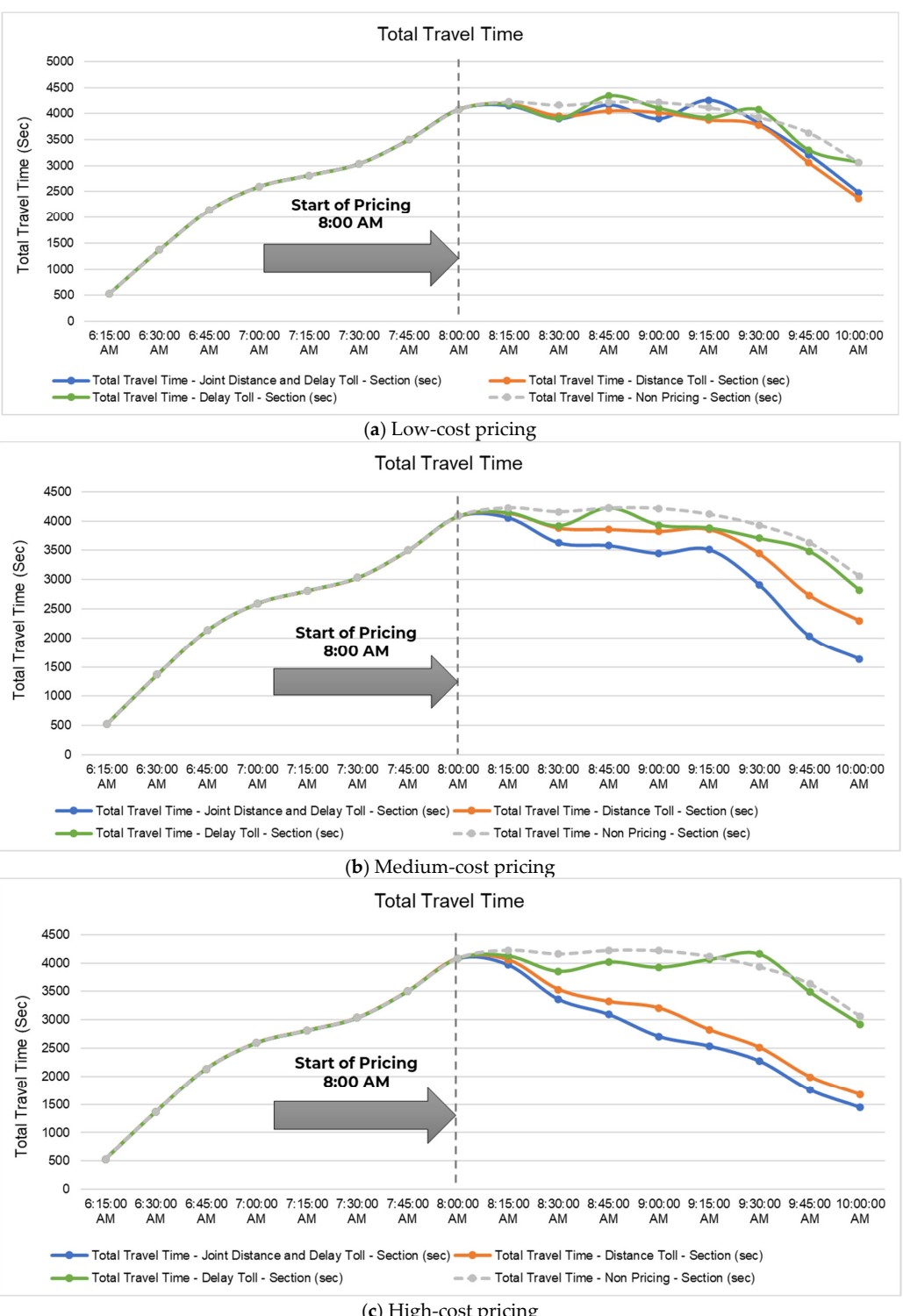

(**a**) Low-cost pricing

(**b**) Medium-cost pricing

(**c**) High-cost pricing

**Figure 7.** Simulated total travel time in pricing zone under different pricing scenarios.

### 4.4. Emissions

The simulation results also include measurements of emissions produced by each vehicle category according to their travel within the network. The emissions model takes into consideration vehicle speed, acceleration, and deceleration to estimate emissions and air pollutants based on established emissions estimation methodologies [34,47,48]. The results revealed that after the start of the pricing period, there was a notable reduction in emissions produced in the network (Figure 8). The emissions performance follows the same pattern as can be seen in other performance indicators, i.e., the high-cost scenario is performing the best of all modeled scenarios. In the same way, the joint-distance and delay-based pricing schemes were best at reducing emissions. Figure 9 compares the carbon emissions within the network in pricing and non-pricing scenarios. It can be seen that in the pricing scenarios, there are fewer red-color links (higher emissions). In the pricing scenario, most links are green except for a few yellow-colored links, which are indicative of lower emissions overall.

The levels of emissions also depend on the total number of vehicles in the network. If the network is not congested, the levels of emissions produced will be lower. With the implementation of pricing, emissions go down with a reduction in traffic count. The maximum reduction can be seen in the high-cost scenario, which is calculated at 13%. A similar study conducted in London using AIMSUN found a reduction in nitrogen and carbon emissions with the implementation of pricing, resulting in a 22% reduction in emissions using a cordon pricing scheme [33].

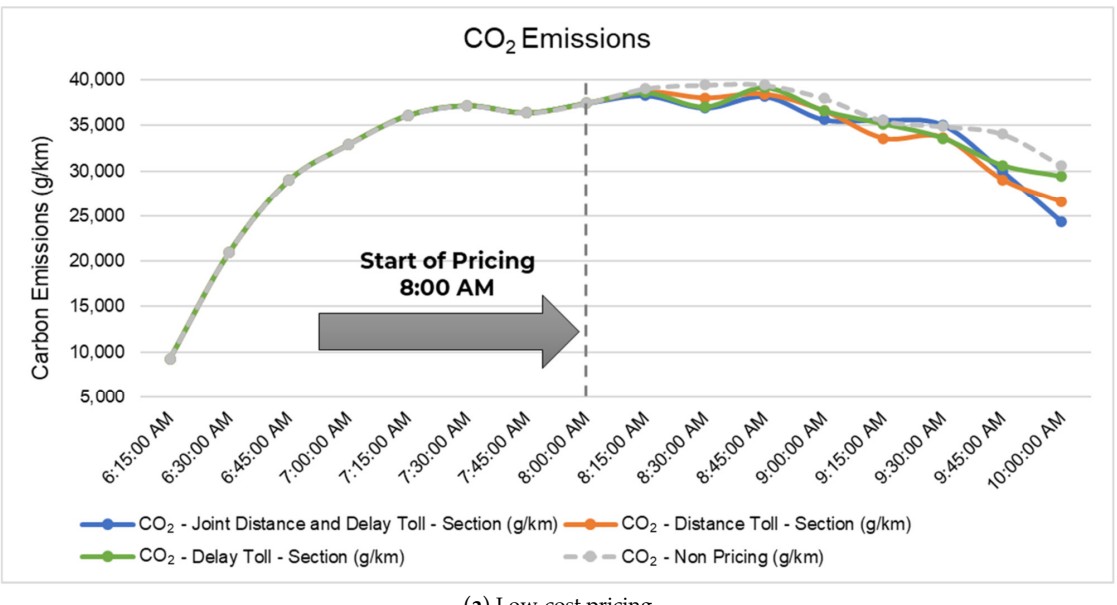

(**a**) Low-cost pricing

**Figure 8.** *Cont.*

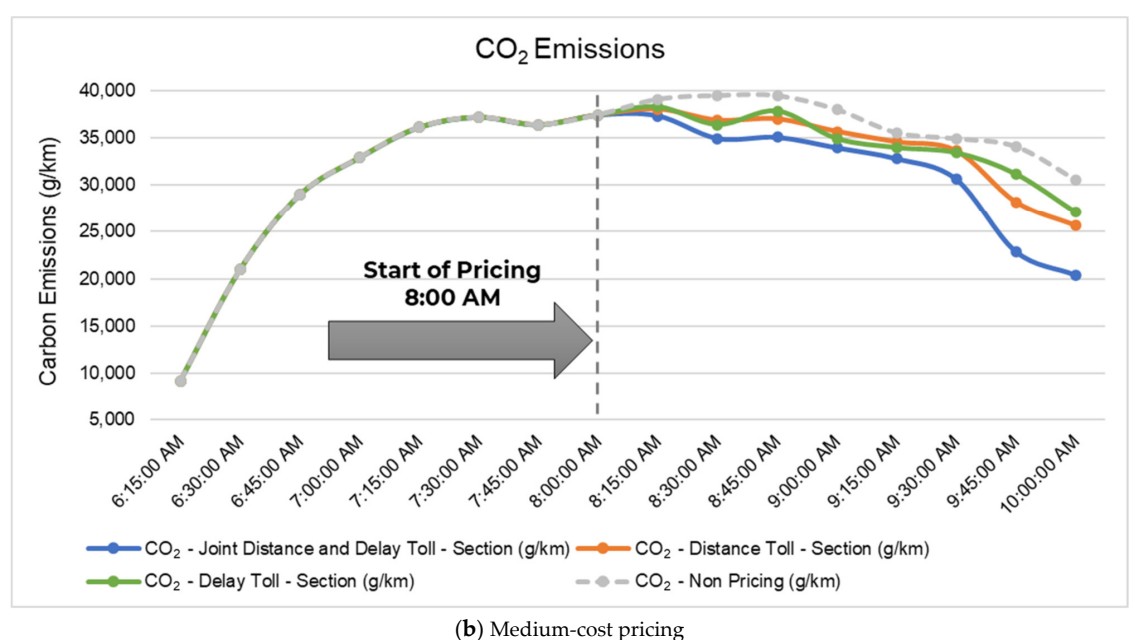

(**b**) Medium-cost pricing

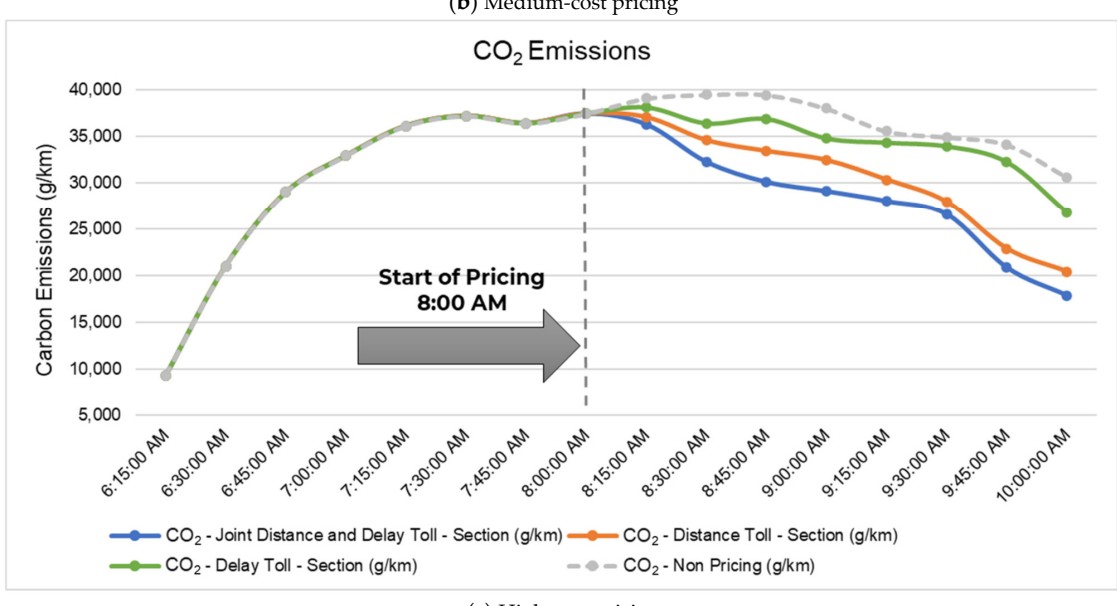

(**c**) High-cost pricing

**Figure 8.** Simulated emissions within the pricing zones under different pricing scenarios.

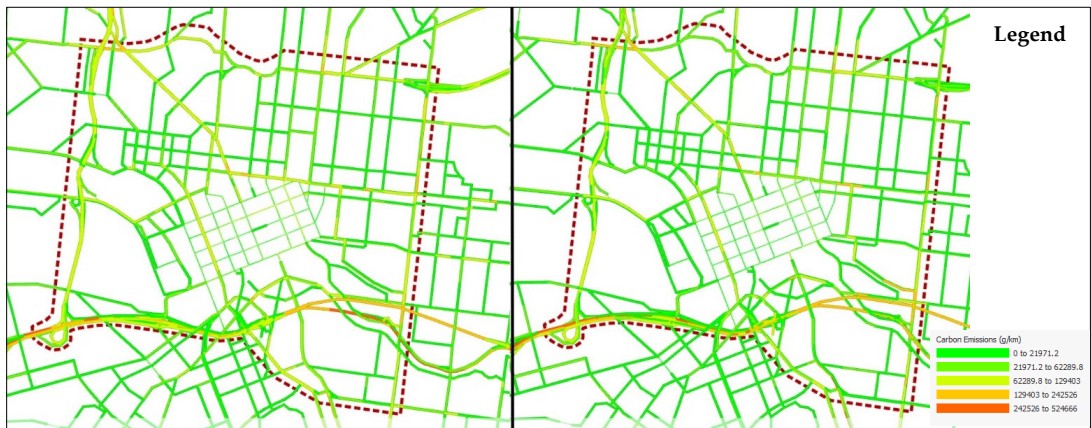

**Figure 9.** Comparison of emissions within the pricing zone under (**right**) pricing and (**left**) non-pricing scenarios.

### 4.5. Travel Speeds

Travel speeds are also key indicators of network performance. In a network where traffic is flowing smoothly, travel speeds tend to be uniform within the designated speed limits. In this analysis, it was found that speeds increased to a notable level in the pricing scenario, within specified speed limits, compared to the non-pricing scenario. As noted in Figure 10, the travel speeds are highest in the high-cost scenario. On the other hand, the joint scheme is also performing well in increasing travel speeds. At a maximum level, it was possible to achieve an increase of 3% in travel speeds in the high-cost scenario, which is equivalent to 3 km/h. In a study that examined similar pricing solutions for future scenarios of autonomous vehicles [43], the researchers achieved a similar increase of 3% in mixed traffic scenarios comprising traditional and autonomous vehicles. Figure 11 provides a visual comparison of travel speeds between the pricing and non-pricing scenarios. It is noted in the diagram how some links that experienced heavy congestion in the baseline non-pricing scenario (links in red) experienced improvements under the pricing scenarios, as evident by the same links turning green, signifying less congested conditions under the pricing scenario.

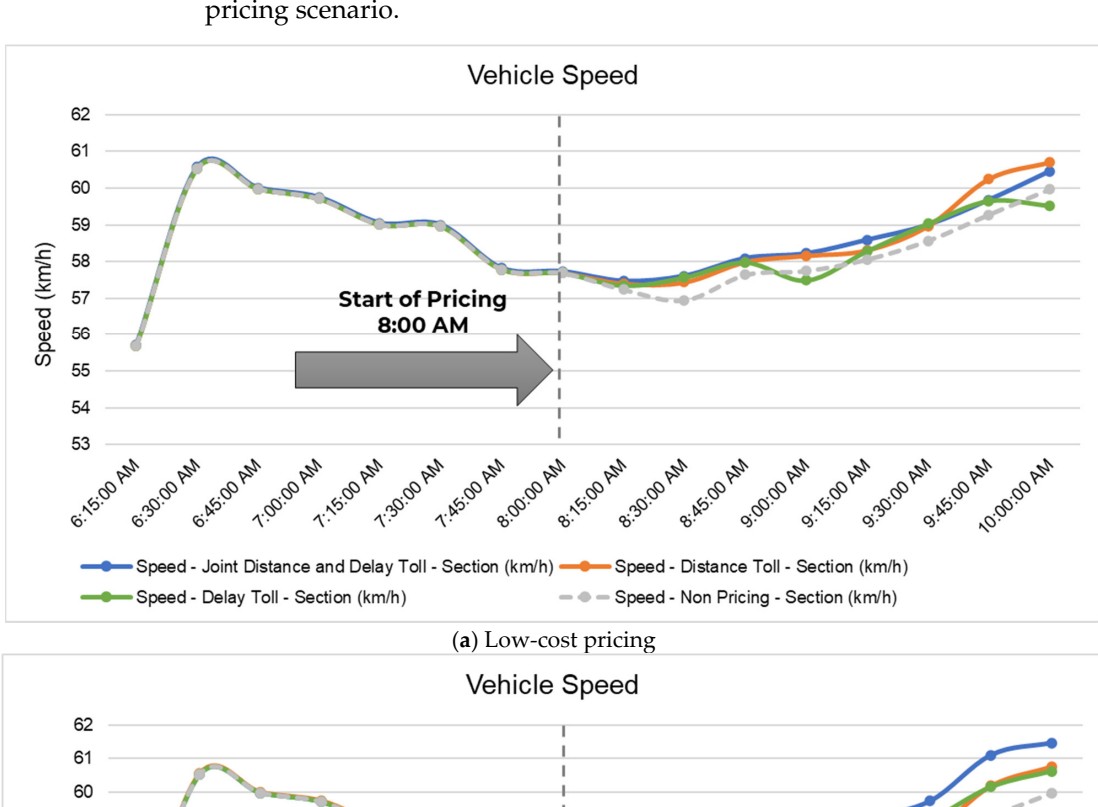

(**a**) Low-cost pricing

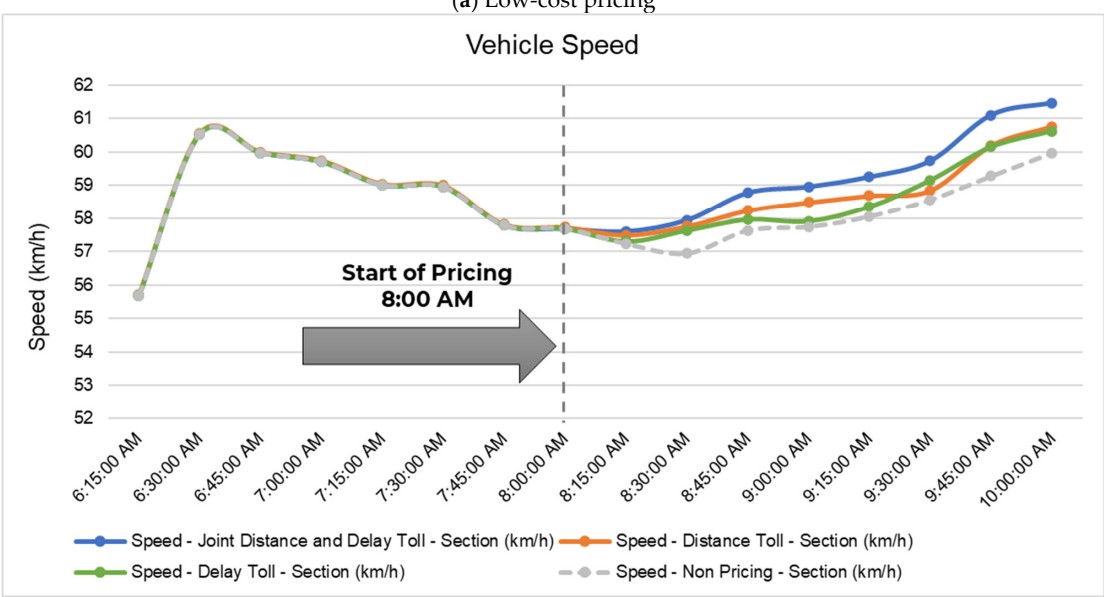

(**b**) Medium-cost pricing

**Figure 10.** *Cont.*

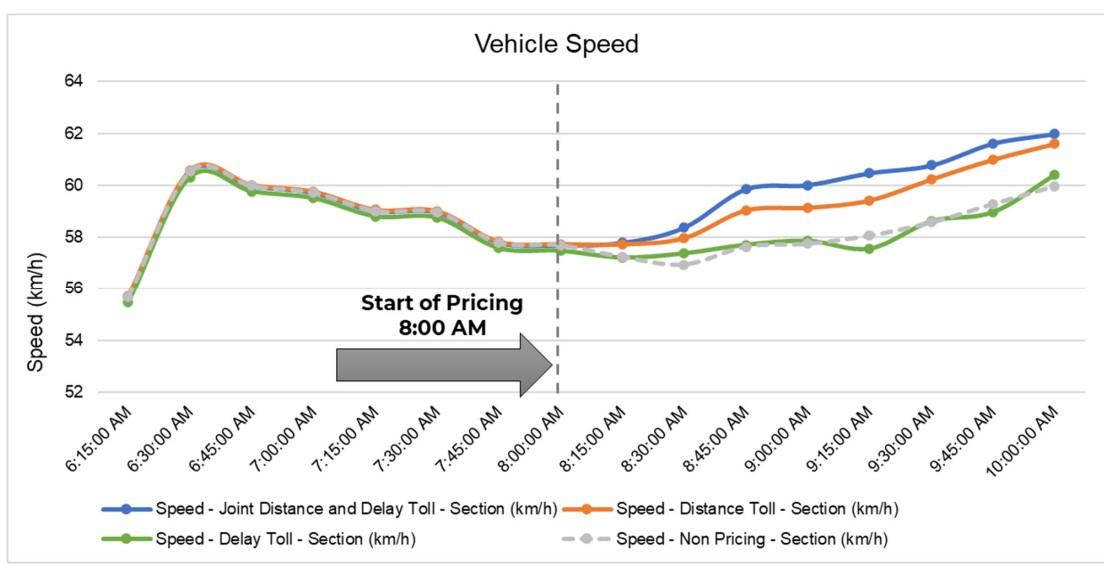

(**c**) High-cost pricing

**Figure 10.** Simulated travel speeds within the pricing zones under different pricing scenarios.

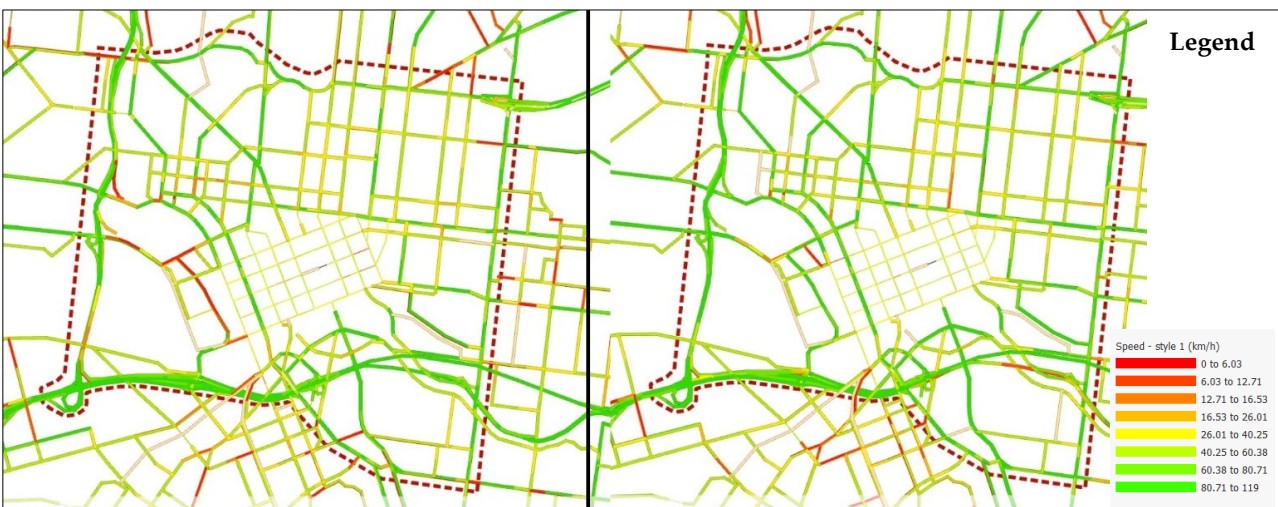

**Figure 11.** Comparison of travel speeds for (**right**) pricing and (**left**) non-pricing scenarios.

*4.6. Comparison of Low-, Medium-, and High-Cost Pricing*

Table 3 and Figure 12 provide a comparison of network performance for the different scenarios and report the percentage improvements of each indicator compared to the baseline non-pricing scenario.

**Table 3.** Network performance in different scenarios.

| Comparison of Network Performance in Different Scenarios—Percentage Improvements over the Baseline Non-Pricing Scenario | | | |
|---|---|---|---|
| Network Performance Indicator | Low-Cost Scenario (%) | Medium-Cost Scenario (%) | High-Cost Scenario (%) |
| Vehicle Count | 3 | 6 | 11 |
| Vehicle Speed | 0.5 | 1 | 3 |
| Total Distance Traveled | 3 | 7 | 11 |
| Total Travel Time | 4 | 15 | 20 |
| $CO_2$ Emissions | 3 | 8 | 13 |
| NOx Emissions | 3 | 8 | 13 |

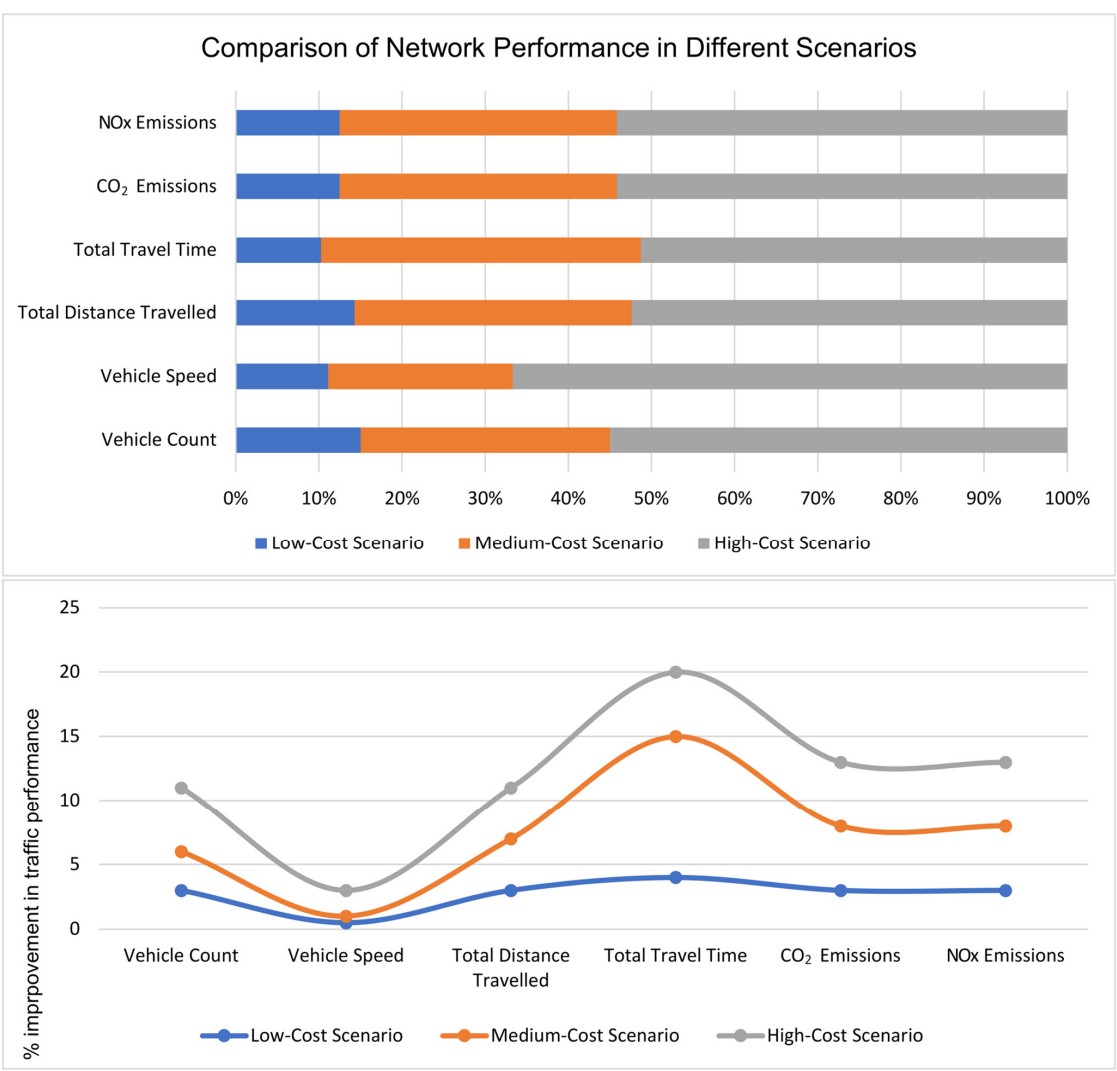

**Figure 12.** Comparison of low-, medium-, and high-cost scenarios.

As would be expected, the high-cost pricing scheme performed best on all indicators but would come at an expense to travelers who choose to drive their vehicles during congested periods. However, in this pricing scheme, the charging rates are set to values that are slightly higher than the value of time for travelers. As such, only travelers whose value of travel time was higher than the charge would take the priced routes. Otherwise, travelers may choose to take alternate, non-priced routes to avoid the charge. As a result, the network performance is highest in this scenario.

In the high-cost scenario, it is observed that there were notable improvements in all traffic performance indicators, including a 20% improvement in travel time. When the network or priced zone has lower vehicle counts, this leads to smoother traffic flow and hence reduced travel time. The travel speed improvements were modest at 3%, largely because most roads in the pricing zone had a speed limit of 60 km/h and speeds in non-pricing scenarios in the same area had already ranged from 55–59 km/h. Although speeds had improved in the pricing scenarios, it was not a significant increase because travelers cannot travel beyond 60 km/h.

Similarly, traffic performance in the medium-cost scenario also improved, but it was slightly lower than in the high-cost scenario. The low-cost scenario also improved traffic performance in the network, but the improvements were minimal because most travelers were willing to pay the small road user charge to avoid diverting from their original route. Overall, and as noted in Figure 12, the traffic performance and improvements all

followed the same pattern in all pricing scenarios, though at different levels of improvement according to the level of pricing. This is an important consideration for policymakers to consider when deciding pricing based on evidence and such that it is realistic and acceptable to the public.

### 4.7. Relaxing Inelastic Transport Demand—Public Transport Scenario

Transport demand is inelastic, especially in the extreme peak hours, and cannot be reduced or suppressed without major behavioral changes or demand management techniques, such as flexible working or urban planning policies, that encourage densification and reduce the demand for travel. At the same time, it is crucial that travelers are provided with different transport mode options, including public transport and active transport. The model, in previous modeling results, included only private vehicle trips and assumed that all the travelers in this network have the same travel preferences for route choices and willingness to pay, which is not a true representation of the real world. To address this issue, the model was refined and extended for public transport ridership. Realistically, the model should provide travelers with the option of choosing an alternative mode of transport if they do not wish to pay. Second, the model should be introduced to multiclass users according to their different values of time.

To reflect the availability of public transport, a total of 12 scenarios were modeled, assuming different percentages of public transport ridership. The results from these simulations that included public transport options are provided in Figure 13. They show, for example, that by shifting 12% of drivers to public transport, network performance can be improved by up to 40%. Shifting 50% of drivers to public transport would improve network performance by around 70%. The 50% mode shift scenario is clearly a hypothetical scenario that may not be possible to achieve in real life. Therefore, two realistic (but highly ambitious) scenarios that assumed 20% and 30% of drivers would shift to public transport during the pricing period were also modeled. These percentages are considered realistic because they correspond to results from a recent survey of travel behavior that was completed for Melbourne and Sydney, which showed that 20–30% of drivers would be willing to shift to public transport rather than pay a congestion charge. The results of the two scenarios (20% and 30% shift) revealed that network performance improves by up to 45% and 55% in traffic dynamics, respectively, compared to the baseline scenario without any pricing. The 30% mode shift scenario resulted in a reduction of 33% in traffic count, a 32% reduction in total distance traveled, a 55% reduction in travel time, a 39% reduction in emissions, and a 4% increase in travel speeds.

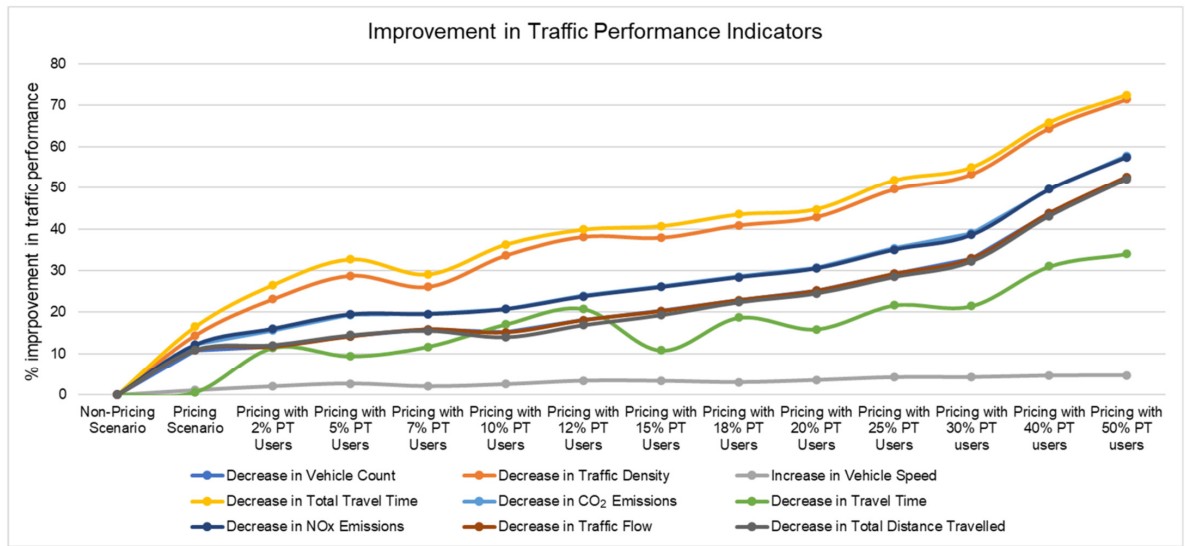

**Figure 13.** Improvements in network performance relative to the percent of drivers who will shift to public transport to avoid paying the congestion charge.

## 5. Conclusions and Future Research Directions

This paper presented findings from a research investigation into the role and possible impacts of road network pricing as an effective travel demand management strategy to reduce traffic congestion and emissions in the context of Melbourne, Australia. In selecting the study and simulation area, it was ensured that there are adequate alternative non-priced routes for travelers who do not wish to pay the road user charges so that they have access to alternative non-priced routes. The papers also explored the impact of road pricing on key traffic performance indicators, i.e., travel time, travel speed, total distance traveled, and maximum queue length. The study relied on a simulation-based methodology using a dynamic traffic simulation model for Melbourne to test various road pricing scenarios, including distance-based pricing, time-based pricing, and a combination of both, under the assumptions of low-cost, medium-cost, and high-cost pricing. The AIMSUN simulation tool was extended in this research for modeling road user pricing and congestion charging, including considerations and formulations of distance-based, delay-based, joint-distance-and-delay-based, and cordon-based schemes under low-cost, medium-cost, and high-cost regimes. The study's contributions also included an extension of the modeling framework to include public transport options to provide travelers with the option of choosing an alternative mode of transport if they do not wish to pay.

The results showed that different pricing strategies had varying impacts on traffic performance. The results showed it would be possible to achieve a reduction of 11% in vehicle count, a 20% reduction in travel time, a 13% reduction in emissions, and a 3% increase in travel speed within the proposed pricing zone under the high-cost pricing scenario. A comparison of the different scenarios showed that joint distance- and time-based pricing, which charges drivers based on the distance they have traveled as well as the time they have spent in the charging zone, provides the best overall benefits.

The results also showed that by shifting 12% of drivers to public transport, network performance can be improved by up to 40%. Under a hypothetical scenario of shifting 50% of drivers to public transport (which is challenging to achieve), this would improve network performance by 70%. Two realistic (but highly ambitious) scenarios that assumed 20% and 30% of drivers would shift to public transport during the pricing period were also modeled, resulting in a 45% and 55% improvement in network performance, respectively.

This research provides opportunities to address key transport challenges facing the world's cities by introducing innovative solutions in road user pricing and focusing on demand management rather than supplying new roads. The findings of this research provide important directions for policymakers in deciding on the type and scope of charging schemes to use and how these could reshape transport taxation by moving away from taxes on vehicles through registration fees and towards user-pay taxations where travelers pay for the travel they do or the pollution and emissions they are responsible for.

Future research should consider modeling other transport modes that can provide travelers with alternatives, such as mobility-as-a-service solutions, and serving the network with sustainable modes, such as active transport, walking, cycling, e-bikes, and e-scooters, in addition to public transport. There is also a need for further research around user heterogeneity by considering multiclass users and differentiating users according to their socioeconomic characteristics and willingness to pay for road user charges versus willingness to shift to other modes of transport to avoid paying a congestion charge. Future research should consider a much larger simulation window covering the full day and applying variable costs of pricing for different durations, e.g., a low price for 2 a.m.–6 a.m., a high price between 6 a.m.–10 a.m. and 4 p.m.–8 p.m., and medium prices at other times of the day. The simulation model can be further updated and extended to include modeling of electric vehicles, highly automated, autonomous, and shared autonomous vehicles to explore the impacts of pricing in a future era when these vehicles make up higher proportions of the vehicle fleet.



**Author Contributions:** Planning and conceptualization: H.D., H.G. and T.M.; conducting the literature review: T.M.; initial drafting of paper contents: T.M. and H.D.; methodology and modelling framework: H.D., T.M., S.S. and H.G.; paper structuring: H.D. and T.M.; writing: T.M. and H.D.; editing: H.D., H.G. and S.S.; proof-reading: T.M., H.D., H.G. and S.S.; analysis: T.M., H.D. and S.S.; curation: T.M.; supervision and mentoring of PhD student: H.D., H.G. and S.S. All authors have read and agreed to the published version of the manuscript.

**Funding:** The authors acknowledge the financial support received for the first author in the form of a PhD scholarship from Swinburne University and the Government of Pakistan under Grant [5-1/HRD/UESTPI(Batch-VI)/8093/2019/HEC]. This research is also partly funded in the form of a PhD scholarship top-up for the first author provided by the iMOVE CRC and supported by the Cooperative Research Centre's program, an Australian Government initiative.

**Institutional Review Board Statement:** Not applicable.

**Informed Consent Statement:** Not applicable.

**Data Availability Statement:** The aggregated data presented in this study are available on request from the corresponding author. The data are not publicly available due to privacy and research ethics clearance requirements.

**Conflicts of Interest:** The authors declare that there are no known conflict of interest.

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
