# Peer review of "Comparative Evaluation of Road Pricing Schemes: A Simulation Approach (Australian Perspective)"

_sustainability, doi:10.3390/su152316366_

Round 1
Reviewer 1 Report
Comments and Suggestions for Authors
The authors are advised to emphasize the significance of their research in the abstract and introduction sections. In this sense, my main concern relies on how this study diverges from others. As evidenced in the state of the art, road pricing schemes have been deeply analyzed and even deployed, shedding light on similar results as those presented in this work.
Equations 1 to 6 are hard to understand mainly due to the employed notation and careless presentation. For instance, why do notations “rs” and “a,k” have different meanings when relating pairs? Or what does theta mean?
The author should pay meticulous attention to the details in the manuscript. There are numerous typos all along the document, for instance, “Weidman” or “tripes” in Section 3.1. In addition, several erroneous references and graphs with strange notation or a lack of appropriate units are found.
Can the authors clarify how traffic authorities can specifically benefit from the findings of this study?
Comments on the Quality of English LanguageThere are numerous typos all along the document.
Author Response
Please see attached word document

Reviewer 2 Report
Comments and Suggestions for Authors
The submitted article is an interesting approach to a relevant investigation area as sustainable transportation and launches clues using an approach to build scenarios correlating several variables: CO2 emissions and pricing.
In the bibliography analysis, it would be interesting to have a summary table comparing the described methodologies, considering results and accuracy.
Be aware that the legend and images should be on the same page.
Figure 5 has some non-translated characters on X-axis.
Several figures don't have units on the Y-axis.
It would be interesting to add a new scenario of the current situation and identify how accurate is the model.
Conclusions should be clear considering the original contribution of the authors. It is not clear the real contribution of the presented methodology and how it compares with existing ones.
Author Response
Please see attached word document

Reviewer 3 Report
Comments and Suggestions for Authors
In the present manuscript, the authors study the approach for the city of Melbourne in Australia to investigate the potential impacts of road network pricing in reducing private vehicle travel, road congestion and vehicle emissions. However, I will comment on some aspects that the authors should improve and should be highlighted.
- The authors must briefly present the AIMSUN simulation tool.
- on line 250, is it the title of the Figure? This is mentioned since it jumps from line 249 to 250.
- What are the fundamental characteristics of why the authors have chosen the study area?
- How was the Origin-Destination matrix obtained to perform the simulation?
- Are the routes used in the simulation always static? Or dynamics according to traffic load balancing?
- The conclusions must be improved and added to future work.
Comments on the Quality of English LanguageIn the present manuscript, the authors study the approach for the city of Melbourne in Australia to investigate the potential impacts of road network pricing in reducing private vehicle travel, road congestion and vehicle emissions. However, I will comment on some aspects that the authors should improve and should be highlighted.
- The authors must briefly present the AIMSUN simulation tool.
- on line 250, is it the title of the Figure? This is mentioned since it jumps from line 249 to 250.
- What are the fundamental characteristics of why the authors have chosen the study area?
- How was the Origin-Destination matrix obtained to perform the simulation?
- Are the routes used in the simulation always static? Or dynamics according to traffic load balancing?
- The conclusions must be improved and added to future work.
Author Response
Please see attached word document

Reviewer 4 Report
Comments and Suggestions for Authors
The article entitled "Comparative Evaluation of Road Pricing Schemes: A Simulation Approach (Australian Perspective)" considers a simulation approach for the city of Melbourne in Australia to investigate the potential impacts of road network pricing in reducing private vehicle travel, road congestion, and vehicle emissions.
The article is well written and the findings are comprehensive and supported by the conclusions.
Minimal changes are required to the article in order to be published.
- review the references to tables and figures: sometimes the message "Error! Reference source not found" appears.
- some formatting problems need to be corrected.
- insert a legend in figures 3, 6, 9, and 11 that clarifies the meaning of the colors and thickness of the lines (dashed or not dashed).
Author Response
Please see attached word document

Round 2
Reviewer 1 Report
Comments and Suggestions for Authors
The authors have provided a comprehensive discussion regarding my main concerns, expressed in the first round of the revision. However, the document still has details that demerit the quality and formality of the proposal. For instance, in Figure 5 the scale for the morning peak time presents strange symbols following the hours of the sample. In addition, in such a Figure a secondary (right-side) scale appears without any legend stating its meaning.
The document still has erroneous references, so the authors argued that such details were caused due to the editorial's template. Nevertheless, the authors must consider that references are pointers to the literature that supports the stated ideas, therefore, authors are responsible for procuring a proper display for such elements.
Author Response
The authors thank Reviewer #1 for the additional comments on our submission. Our responses are provided below
Reviewer Comments
The authors have provided a comprehensive discussion regarding my main concerns, expressed in the first round of the revision. However, the document still has details that demerit the quality and formality of the proposal.
For instance, in Figure 5 the scale for the morning peak time presents strange symbols following the hours of the sample.
Response
We have checked. We cannot see any strange symbols. We can only see that the times provided had "AM" after them to denote the morning peak (for example 08:00:00 AM). We have now removed this text from Figure 5 and only kept the time stamp.
In addition, in such a Figure a secondary (right-side) scale appears without any legend stating its meaning.
A legend has now been added to the right vertical axis
The document still has erroneous references, so the authors argued that such details were caused due to the editorial's template. Nevertheless, the authors must consider that references are pointers to the literature that supports the stated ideas, therefore, authors are responsible for procuring a proper display for such elements.
We have redone the references using the Vancouver system. When the paper is approved for publication, we will work with the editorial team to ensure that the referencing formats are consistent with journal requirements.
Round 3
Reviewer 1 Report
Comments and Suggestions for Authors
I think that after the authors solve the problems with the citations, the paper could be accepted for publication.
Author Response
The authors would like to thank Reviewer #1 who has now approved the revised manuscript and indicated that the paper can be accepted for publication.
The authors have now refreshed the paper and applied all the accepted changes and uploaded a final version to the portal for the attention of the MDPI editorial team.